# Multi-Step Stochastic ADMM in High Dimensions: Applications to Sparse Optimization and Matrix Decomposition

**Hanie Sedghi**
Univ. of Southern California
Los Angeles, CA 90089
hsedghi@usc.edu

**Anima Anandkumar**
University of California
Irvine, CA 92697
a.anandkumar@uci.edu

**Edmond Jonckheere**
Univ. of Southern California
Los Angeles, CA 90089
jonckhee@usc.edu

## Abstract

In this paper, we consider a multi-step version of the stochastic ADMM method with efficient guarantees for high-dimensional problems. We first analyze the simple setting, where the optimization problem consists of a loss function and a single regularizer (e.g. sparse optimization), and then extend to the multi-block setting with multiple regularizers and multiple variables (e.g. matrix decomposition into sparse and low rank components). For the sparse optimization problem, our method achieves the minimax rate of $O(s \log d/T)$ for $s$-sparse problems in $d$ dimensions in $T$ steps, and is thus, unimprovable by any method up to constant factors. For the matrix decomposition problem with a general loss function, we analyze the multi-step ADMM with multiple blocks. We establish $O(1/T)$ rate and efficient scaling as the size of matrix grows. For natural noise models (e.g. independent noise), our convergence rate is minimax-optimal. Thus, we establish tight convergence guarantees for multi-block ADMM in high dimensions. Experiments show that for both sparse optimization and matrix decomposition problems, our algorithm outperforms the state-of-the-art methods.

## 1 Introduction

Stochastic optimization techniques have been extensively employed for online machine learning on data which is uncertain, noisy or missing. Typically it involves performing a large number of inexpensive iterative updates, making it scalable for large-scale learning. In contrast, traditional batch-based techniques involve far more expensive operations for each update step. Stochastic optimization has been analyzed in a number of recent works.

The alternating direction method of multipliers (ADMM) is a popular method for online and distributed optimization on a large scale [1], and is employed in many applications. It can be viewed as a decomposition procedure where solutions to sub-problems are found locally, and coordinated via constraints to find the global solution. Specifically, it is a form of augmented Lagrangian method which applies partial updates to the dual variables. ADMM is often applied to solve regularized problems, where the function optimization and regularization can be carried out locally, and then coordinated globally via constraints. Regularized optimization problems are especially relevant in the high dimensional regime since regularization is a natural mechanism to overcome ill-posedness and to encourage parsimony in the optimal solution, e.g., sparsity and low rank. Due to the efficiency of ADMM in solving regularized problems, we employ it in this paper.

We consider a simple modification to the (inexact) stochastic ADMM method [2] by incorporating multiple steps or epochs, which can be viewed as a form of annealing. We establish that this simple modification has huge implications in achieving tight bounds on convergence rate as the dimensions

of the problem instances scale. In each iteration, we employ projections on to certain norm balls of appropriate radii, and we decrease the radii in epochs over time. For instance, for the sparse optimization problem, we constrain the optimal solution at each step to be within an $\ell_1$-norm ball of the initial estimate, obtained at the beginning of each epoch. At the end of the epoch, an average is computed and passed on to the next epoch as its initial estimate. Note that the $\ell_1$ projection can be solved efficiently in linear time, and can also be parallelized easily [3]. For matrix decomposition with a general loss function, the ADMM method requires multiple blocks for updating the low rank and sparse components. We apply the same principle and project the sparse and low rank estimates on to $\ell_1$ and nuclear norm balls, and these projections can be computed efficiently.

**Theoretical implications:** The above simple modifications to ADMM have huge implications for high-dimensional problems. For sparse optimization, our convergence rate is $\mathcal{O}(\frac{s \log d}{T})$, for $s$-sparse problems in $d$ dimensions in $T$ steps. Our bound has the best of both worlds: efficient high-dimensional scaling (as $\log d$) and efficient convergence rate (as $\frac{1}{T}$). This also matches the minimax rate for the linear model and square loss function [4], which implies that our guarantee is unimprovable by any (batch or online) algorithm (up to constant factors). For matrix decomposition, our convergence rate is $\mathcal{O}((s+r)\beta^2(p)\log p/T)) + \mathcal{O}(\max\{s+r,p\}/p^2)$ for a $p \times p$ input matrix in $T$ steps, where the sparse part has $s$ non-zero entries and low rank part has rank $r$. For many natural noise models (e.g. independent noise, linear Bayesian networks), $\beta^2(p) = p$, and the resulting convergence rate is minimax-optimal. Note that our bound is not only on the reconstruction error, but also on the error in recovering the sparse and low rank components. These are the first convergence guarantees for online matrix decomposition in high dimensions. Moreover, our convergence rate holds *with high probability* when noisy samples are input, in contrast to expected convergence rate, typically analyzed in the literature. See Table 1, 2 for comparison of this work with related frameworks. Proof of all results and implementation details can be found in the longer version [5].

**Practical implications:** The proposed algorithms provide significantly faster convergence in high dimension and better robustness to noise. For sparse optimization, our method has significantly better accuracy compared to the stochastic ADMM method and better performance than RADAR, based on multi-step dual averaging [6]. For matrix decomposition, we compare our method with the state-of-art inexact ALM [7] method. While both methods have similar reconstruction performance, our method has significantly better accuracy in recovering the sparse and low rank components.

**Related Work: ADMM:** Existing online ADMM-based methods lack high-dimensional guarantees. They scale poorly with the data dimension (as $\mathcal{O}(d^2)$), and also have slow convergence for general problems (as $\mathcal{O}(\frac{1}{\sqrt{T}})$). Under strong convexity, the convergence rate can be improved to $\mathcal{O}(\frac{1}{T})$ but only in *expectation*: such analyses ignore the per sample error and consider only the expected convergence rate(see Table 1). In contrast, our bounds hold with high probability. Some stochastic ADMM methods, Goldstein et al. [8], Deng [9] and Luo [10] provide faster rates for stochastic ADMM, than the rate noted in Table 1. However, they require strong conditions which are not satisfied for the optimization problems considered here, e.g., Goldstein et al. [8] require both the loss function and the regularizer to be strongly convex.

**Related Work: Sparse Optimization:** For the sparse optimization problem, $\ell_1$ regularization is employed and the underlying true parameter is assumed to be sparse. This is a well-studied problem in a number of works (for details, refer to [6]). Agarwal et al. [6] propose an efficient online method based on dual averaging, which achieves the same optimal rates as the ones derived in this paper. The main difference is that our ADMM method is capable of solving the problem for multiple random variables and multiple conditions while their method cannot incorporate these extensions.

**Related Work: Matrix Decomposition:** To the best of our knowledge, online guarantees for high-dimensional matrix decomposition have not been provided before. Wang et al. [12] propose a multi-block ADMM method for the matrix decomposition problem but only provide convergence rate analysis in expectation and it has poor high dimensional scaling (as $\mathcal{O}(p^4)$ for a $p \times p$ matrix) without further modifications. Note that they only provide convergence rate on difference between loss function and optimal loss, whereas we provide the convergence rate on individual errors of the sparse and low rank components $\|\bar{S}(T) - S^*\|_{\mathbb{F}}^2, \|\bar{L}(T) - L^*\|_{\mathbb{F}}^2$. See Table 2 for comparison of guarantees for matrix decomposition problem.

**Notation** In the sequel, we use lower case letter for vectors and upper case letter for matrices. Moreover, $X \in \mathbb{R}^{p \times p}$. $\|x\|_1, \|x\|_2$ refer to $\ell_1, \ell_2$ vector norms respectively. The term $\|X\|_*$ stands

| Method | Assumptions | Convergence rate |
|---|---|---|
| ST-ADMM [2] | L, convexity | $\mathcal{O}(d^2/\sqrt{T})$ |
| ST-ADMM [2] | SC, E | $\mathcal{O}(d^2 \log T/T)$ |
| BADMM [11] | convexity, E | $\mathcal{O}(d^2/\sqrt{T})$ |
| RADAR [6] | LSC, LL | $\mathcal{O}(s \log d/T)$ |
| REASON 1 (this paper) | LSC, LL | $\mathcal{O}(s \log d/T)$ |
| Minimax bound [4] | Eigenvalue conditions | $\mathcal{O}(s \log d/T)$ |

Table 1: *Comparison of online sparse optimization methods under $s$ sparsity level for the optimal paramter, $d$ dimensional space, and $T$ number of iterations. SC = Strong Convexity, LSC = Local Strong Convexity, LL = Local Lipschitz, L = Lipschitz property, E=in Expectation. The last row provides the minimax-optimal rate for any method. The results hold with high probability.*

| Method | Assumptions | Convergence rate |
|---|---|---|
| Multi-block-ADMM[12] | L, SC, E | $\mathcal{O}(p^4/T)$ |
| Batch method[13] | LL, LSC, DF | $\mathcal{O}((s \log p + rp)/T)+\mathcal{O}(s/p^2)$ |
| REASON 2 (this paper) | LSC, LL, DF | $\mathcal{O}((s+r)\beta^2(p) \log p/T))+\mathcal{O}(\max\{s+r,p\}/p^2)$ |
| Minimax bound[13] | $\ell_2$, IN, DF | $\mathcal{O}((s \log p + rp)/T)+\mathcal{O}(s/p^2)$ |

Table 2: *Comparison of optimization methods for sparse+low rank matrix decomposition for a $p \times p$ matrix under $s$ sparsity level and $r$ rank matrices and $T$ is the number of samples. Abbreviations are as in Table 1, IN = Independent noise model, DF = diffuse low rank matrix under the optimal parameter. $\beta(p) = \Omega(\sqrt{p}), \mathcal{O}(p)$ and its value depends the model. The last row provides the minimax-optimal rate for any method under the independent noise model. The results hold with high probability unless otherwise mentioned. For Multi-block-ADMM [12] the convergence rate is on the difference of loss function from optimal loss, for the rest of works in the table, the convergence rate is on the individual estimates of the sparse and low rank components: $\|\bar{S}(T) - S^*\|_{\mathbb{F}}^2 + \|\bar{L}(T) - L^*\|_{\mathbb{F}}^2$.* for nuclear norm of $X$. In addition, $\|X\|_2, \|X\|_{\mathbb{F}}$ denote spectral and Frobenius norms respectively. We use vectorized $\ell_1, \ell_\infty$ norm for matrices, i.e., $\|X\|_1 = \sum_{i,j}|X_{ij}|, \|X\|_\infty = \max_{i,j}|X_{ij}|$.

## 2 $\ell_1$ Regularized Stochastic Optimization

We consider the optimization problem $\theta^* \in \arg\min \mathbb{E}[f(\theta, x)], \theta \in \Omega$ where $\theta^*$ is a sparse vector. The loss function $f(\theta, x_k)$ is a function of a parameter $\theta \in \mathbb{R}^d$ and samples $x_i$. In stochastic setting, we do not have access to $\mathbb{E}[f(\theta, x)]$ nor to its subgradients. In each iteration we have access to one noisy sample. In order to impose sparsity we use regularization. Thus, we solve a sequence

$$\theta_k \in \arg\min_{\theta \in \Omega'} f(\theta, x_k) + \lambda\|\theta\|_1, \quad \Omega' \subset \Omega, \qquad (1)$$

where the regularization parameter $\lambda > 0$ and the constraint sets $\Omega'$ change from epoch to epoch.

### 2.1 Epoch-based Stochastic ADMM Algorithm

We now describe the modified inexact ADMM algorithm for the sparse optimization problem in (1), and refer to it as REASON 1, see Algorithm 1. We consider an epoch length $T_0$, and in each epoch $i$, we project the optimal solution on to an $\ell_1$ ball with radius $R_i$ centered around $\tilde{\theta}_i$, which is the initial estimate of $\theta^*$ at the start of the epoch. The $\theta$-update is given by

$$\theta_{k+1} = \arg\min_{\|\theta - \tilde{\theta}_i\|_1^2 \leq R_i^2} \{\langle \nabla f(\theta_k), \theta - \theta_k \rangle - \langle z_k, \theta - y_k \rangle + \frac{\rho}{2}\|\theta - y_k\|_2^2 + \frac{\rho_x}{2}\|\theta - \theta_k\|_2^2\}. \qquad (2)$$

Note that this is an inexact update since we employ the gradient $\nabla f(\cdot)$ rather than optimize directly on the loss function $f(\cdot)$ which is expensive. The above program can be solved efficiently since it is a projection on to the $\ell_1$ ball, whose complexity is linear in the sparsity level of the gradient, when performed serially, and $\mathcal{O}(\log d)$ when performed in parallel using $d$ processors [3]. For the regularizer, we introduce the variable $y$, and the $y$-update is $y_{k+1} = \arg\min\{\lambda_i\|y_k\|_1 - \langle z_k, \theta_{k+1} -$

---

**Algorithm 1:** Regularized Epoch-based Admm for Stochastic Opt. in high-dimensioN 1 (REASON 1)

---

**Input** $\rho, \rho_x$, epoch length $T_0$ , initial prox center $\tilde{\theta}_1$, initial radius $R_1$, regularization parameter $\{\lambda_i\}_{i=1}^{k_T}$.

**Define** $\text{Shrink}_\kappa(a) = (a - \kappa)_+ - (-a - \kappa)_+$.

**for** *Each epoch* $i = 1, 2, ..., k_T$ **do**

  Initialize $\theta_0 = y_0 = \tilde{\theta}_i$

  **for** *Each iteration* $k = 0, 1, ..., T_0 - 1$ **do**

$$\theta_{k+1} = \underset{\|\theta - \tilde{\theta}_i\|_1 \le R_i}{\arg\min} \{\langle \nabla f(\theta_k), \theta - \theta_k \rangle - \langle z_k, \theta - y_k \rangle + \frac{\rho}{2}\|\theta - y_k\|_2^2 + \frac{\rho_x}{2}\|\theta - \theta_k\|_2^2\}$$

$$y_{k+1} = \text{Shrink}_{\lambda_i/\rho}(\theta_{k+1} - \frac{z_k}{\rho}), \qquad z_{k+1} = z_k - \tau(\theta_{k+1} - y_{k+1})$$

  **Return** : $\overline{\theta}(T_i) := \frac{1}{T}\sum_{k=0}^{T_0-1} \theta_k$ for epoch $i$ and $\tilde{\theta}_{i+1} = \overline{\theta}(T_i)$.

  **Update** : $R_{i+1}^2 = R_i^2/2$.

---

$y\rangle + \frac{\rho}{2}\|\theta_{k+1} - y\|_2^2\}$. This update can be simplified to the form given in REASON 1, where $\text{Shrink}_\kappa(\cdot)$ is the soft-thresholding or shrinkage function [1]. Thus, each step in the update is extremely simple to implement. When an epoch is complete, we carry over the average $\overline{\theta}(T_i)$ as the next epoch center and reset the other variables.

## 2.2 High-dimensional Guarantees

We now provide convergence guarantees for the proposed method under the following assumptions.

**Assumption A1: Local strong convexity (LSC)**: The function $f : S \to R$ satisfies an $R$-local form of strong convexity (LSC) if there is a non-negative constant $\gamma = \gamma(R)$ such that for any $\theta_1, \theta_2 \in S$ with $\|\theta_1\|_1 \le R$ and $\|\theta_2\|_1 \le R$, $f(\theta_1) \ge f(\theta_2) + \langle \nabla f(\theta_2), \theta_1 - \theta_2 \rangle + \frac{\gamma}{2}\|\theta_2 - \theta_1\|_2^2$.

Note that the notion of strong convexity leads to faster convergence rates in general. Intuitively, strong convexity is a measure of curvature of the loss function, which relates the reduction in the loss function to closeness in the variable domain. Assuming that the function $f$ is twice continuously differentiable, it is strongly convex, if and only if its Hessian is positive semi-definite, for all feasible $\theta$. However, in the high-dimensional regime, where there are fewer samples than data dimension, the Hessian matrix is often singular and we do not have global strong convexity. A solution is to impose local strong convexity which allows us to provide guarantees for high dimensional problems. This notion has been exploited before in a number of works on high dimensional analysis, e.g., [14, 13, 6]. It holds for various loss functions such as square loss.

**Assumption A2: Sub-Gaussian stochastic gradients**: Let $e_k(\theta) := \nabla f(\theta, x_k) - \mathbb{E}[\nabla f(\theta, x_k)]$. There is a constant $\sigma = \sigma(R)$ such that for all $k > 0$, $\mathbb{E}[\exp(\|e_k(\theta)\|_\infty^2)/\sigma^2] \le \exp(1)$, for all $\theta$ such that $\|\theta - \theta^*\|_1 \le R$.

**Remark:** The bound holds with $\sigma = \mathcal{O}(\sqrt{\log d})$ whenever each component of the error vector has sub-Gaussian tails [6].

**Assumption A3: Local Lipschitz condition**: For each $R > 0$, there is a constant $G = G(R)$ such that, $|f(\theta_1) - f(\theta_2)| \le G\|\theta_1 - \theta_2\|_1$, for all $\theta_1, \theta_2 \in S$ such that $\|\theta - \theta^*\|_1 \le R$ and $\|\theta_1 - \theta^*\|_1 \le R$.

The design parameters are as below where $\lambda_i$ is the regularization for $\ell_1$ term in epoch $i$, $\rho$ and $\rho_x$ are penalties in $\theta$-update as in (2) and $\tau$ is the step size for the dual update.

$$\lambda_i^2 = \frac{\gamma R_i}{s\sqrt{T_0}}\sqrt{\log d + \frac{G^2(\rho + \rho_x)^2}{T_0^2} + \sigma_i^2 \log(\frac{3}{\delta_i})}, \quad \rho \propto \frac{\sqrt{T_0 \log d}}{R_i}, \quad \rho_x > 0, \quad \tau \propto \frac{\sqrt{T_0}}{R_i}.$$
$$(3)$$

**Theorem 1.** *Under Assumptions* $A1 - A3$, $\lambda_i$ *as in* (3) *, with fixed epoch lengths* $T_0 = T \log d/k_T$, *where $T$ is the total number of iterations and*

$$k_T = \log_2 \frac{\gamma^2 R_1^2 T}{s^2(\log d + \frac{\gamma}{s}G + 12\sigma^2 \log(\frac{6}{\delta}))},$$

*and $T_0$ satisfies $T_0 = \mathcal{O}(\log d)$, for any $\theta^*$ with sparsity $s$, with probability at least $1 - \delta$ we have*

$$\|\bar{\theta}_T - \theta^*\|_2^2 = \mathcal{O}\left( s\, \frac{\log d + \frac{\gamma}{s}G + (\log(1/\delta) + \log(k_T/\log d))\sigma^2}{T}\, \frac{\log d}{k_T} \right),$$

*where $\bar{\theta}_T$ is the average for the last epoch for a total of $T$ iterations.*

**Improvement of $\log d$ factor :** The above theorem covers the practical case where the epoch length $T_0$ is fixed. We can improve the above results using varying epoch length (which depend on the problem parameters) such that $\|\bar{\theta}_T - \theta^*\|_2^2 = \mathcal{O}(s \log d/T)$. The details can be found in the longer version [5].This convergence rate of $\mathcal{O}(s \log d/T)$ matches the minimax lower bounds for sparse estimation [4]. This implies that our guarantees are *unimprovable* up to constant factors.

# 3 Extension to Doubly Regularized Stochastic Optimization

We consider the optimization problem $M^* \in \arg\min \mathbb{E}[f(M, X)]$, where we want to decompose $M$ into a sparse matrix $S \in \mathbb{R}^{p \times p}$ and a low rank matrix $L \in \mathbb{R}^{p \times p}$. $f(M, X_k)$ is a function of a parameter $M$ and samples $X_k$. $X_k$ can be a matrix (e.g. independent noise model) or a vector (e.g. Gaussian graphical model). In stochastic setting, we do not have access to $\mathbb{E}[f(M, X)]$ nor to its subgradients. In each iteration, we have access to one noisy sample and update our estimate based on that. We impose the desired properties with regularization. Thus, we solve a sequence

$$\widehat{M}_k := \arg\min\{\widehat{f}(M, X_k) + \lambda_n \|S\|_1 + \mu_n \|L\|_*\} \qquad s.t. \quad M = S + L, \quad \|L\|_\infty \leq \frac{\alpha}{p}. \quad (4)$$

We propose an online program based on multi-block ADMM algorithm. In addition to tailoring projection ideas employed for sparse case, we impose an $\ell_\infty$ constraint of $\alpha/p$ on each entry of $L$. This constraint is also imposed for the batch version of the problem (4) in [13], and we assume that the true matrix $L^*$ satisfies this constraint. Intuitively, the $\ell_\infty$ constraint controls the "spikiness" of $L^*$. If $\alpha \approx 1$, then the entries of $L$ are $\mathcal{O}(1/p)$, i.e. they are "diffuse" or "non-spiky", and no entry is too large. When the low rank matrix $L^*$ has diffuse entries, it cannot be a sparse matrix, and thus, can be separated from the sparse $S^*$ efficiently. In fact, the $\ell_\infty$ constraint is a weaker form of the *incoherence*-type assumptions needed to guarantee identifiability [15] for sparse+low rank decomposition. For more discussions, see Section 3.2.

## 3.1 Epoch-based Multi-Block ADMM Algorithm

We now extend the ADMM method proposed in REASON 1 to multi-block ADMM. The details are in Algorithm 2, and we refer to it as REASON 2. Recall that the matrix decomposition setting assumes that the true matrix $M^* = S^* + L^*$ is a combination of a sparse matrix $S^*$ and a low rank matrix $L^*$. In REASON 2, the updates for matrices $M, S, L$ are done independently at each step. The updates follow definition of ADMM and ideas presented in Section 2. We consider epochs of lengths $T_0$. We do not need to project the update of matrix $M$. The update rules for $S$, $L$ are result of doing an inexact proximal update by considering them as a single block, which can then be decoupled. We impose an $\ell_1$-norm projection for the sparse estimate $S$ around the epoch initialization $\tilde{S}_i$. For the low rank estimate $L$, we impose a nuclear norm projection around the epoch initialization $\tilde{L}_i$. Intuitively, the nuclear norm projection, which is an $\ell_1$ projection on the singular values, encourages sparsity in the spectral domain leading to low rank estimates. We also require an $\ell_\infty$ constraint on $L$. Thus, the update rule for $L$ has two projections, i.e. infinity and nuclear norm projections. We decouple it into ADMM updates $L, Y$ with dual variable $U$ corresponding to this decomposition.

## 3.2 High-dimensional Guarantees

We now prove that REASON 2 recovers both the sparse and low rank estimates in high dimensions efficiently. We need the following assumptions, in addition to Assumptions A2, A3.

**Assumption A4: Spectral Bound on the Gradient Error** Let $E_k(M, X_k) := \nabla f(M, X_k) - \mathbb{E}[\nabla f(M, X_k)]$, $\|E_k\|_2 \leq \beta(p)\sigma$, where $\sigma := \|E_k\|_\infty$.

Recall from Assumption A2 that $\sigma = \mathcal{O}(\log p)$, under sub-Gaussianity. Here, we require spectral bounds in addition to $\|\cdot\|_\infty$ bound in A2.

**Assumption A5: Bound on spikiness of low-rank matrix** $\|L^*\|_\infty \leq \frac{\alpha}{p}$, as discussed before.

**Assumption A6: Local strong convexity (LSC)** The function $f : \mathbb{R}^{d_1 \times d_2} \to \mathbb{R}^{n_1 \times n_2}$ satisfies an $R$-local form of strong convexity (LSC) if there is a non-negative constant $\gamma = \gamma(R)$ such that $f(B_1) \geq f(B_2) + \text{Tr}(\nabla f(B_2)(B_1 - B_2)) + \frac{\gamma}{2}\|B_2 - B_1\|_{\mathbb{F}}$, for any $\|B_1\| \leq R$ and $\|B_2\| \leq R$, which is essentially the matrix version of Assumption A1.

We choose algorithm parameters as below where $\lambda_i, \mu_i$ are the regularization for $\ell_1$ and nuclear norm respectively, $\rho, \rho_x$ correspond to penalty terms in $M$-update and $\tau$ is dual update step size.

$$\lambda_i^2 = \frac{\gamma\sqrt{(R_i^2 + \tilde{R}_i^2)}}{(s+r)\sqrt{T_0}}\sqrt{\log p + \frac{G^2(\rho + \rho_x)^2}{T_0^2} + \beta^2(p)\sigma_i^2 \log(\frac{3}{\delta_i}) + \frac{\alpha^2}{p^2} + \frac{\beta^2(p)\sigma^2}{T_0}\left(\log p + \log\frac{1}{\delta}\right)}$$

(5)

$$\mu_i^2 = c_\mu \lambda_i^2, \quad \rho \propto \sqrt{\frac{T_0 \log p}{R_i^2 + \tilde{R}_i^2}}, \quad \rho_x > 0, \quad \tau \propto \sqrt{\frac{T_0}{R_i^2 + \tilde{R}_i^2}}$$

**Theorem 2.** *Under Assumptions $A2 - A6$, parameter settings* (5)*, let $T$ denote total number of iterations and $T_0 = T \log p / k_T$, where*

$$k_T \simeq -\log\left(\frac{(s+r)^2}{\gamma^2 R_1^2 T}\left[\log p + \frac{G}{s+r} + \beta^2(p)\sigma^2\left[(1+G)(\log(6/\delta) + \log k_T) + \log p\right]\right]\right),$$

*and $T_0$ satisfies $T_0 = \mathcal{O}(\log p)$, with probability at least $1 - \delta$ we have*

$$\|\bar{S}(T) - S^*\|_{\mathbb{F}}^2 + \|\bar{L}(T) - L^*\|_{\mathbb{F}}^2 =$$

$$\mathcal{O}\left((s+r)\frac{\log p + G + \beta^2(p)\sigma^2\left[(1+G)(\log\frac{6}{\delta} + \log\frac{k_T}{\log p}) + \log p\right]}{T}\frac{\log p}{k_T}\right) + \left(1 + \frac{s+r}{\gamma^2 p}\right)\frac{\alpha^2}{p}.$$

**Improvement of $\log p$ factor :** The above result can be improved by a $\log p$ factor by considering varying epoch lengths (which depend on problem parameters). The resulting convergence rate is $\mathcal{O}((s+r)p\log p/T + \alpha^2/p)$. The details can be found in the longer version [5].

**Scaling of $\beta(p)$:** We have the following bounds $\Theta(\sqrt{p}) \leq \beta(p)\Theta(p)$. This implies that the convergence rate (with varying epoch lengths) is $\mathcal{O}((s+r)p\log p/T + \alpha^2/p)$, when $\beta(p) = \Theta(\sqrt{p})$ and when $\beta(p) = \Theta(p)$, it is $\mathcal{O}((s+r)p^2\log p/T + \alpha^2/p)$. The upper bound on $\beta(p)$ arises trivially by converting the max-norm $\|E_k\|_\infty \leq \sigma$ to the bound on the spectral norm $\|E_k\|_2$. In many interesting scenarios, the lower bound on $\beta(p)$ is achieved, as outlined below in Section 3.2.1.

**Comparison with the batch result:** Agarwal et al. [13] consider the batch version of the same problem (4), and provide a convergence rate of $\mathcal{O}((s\log p + rp)/T + s\alpha^2/p^2)$. This is also the minimax lower bound under the independent noise model. With respect to the convergence rate, we match their results with respect to the scaling of $s$ and $r$, and also obtain a $1/T$ rate. We match the scaling with respect to $p$ (up to a log factor), when $\beta(p) = \Theta(\sqrt{p})$ attains the lower bound, and we discuss a few such instances below. Otherwise, we are worse by a factor of $p$ compared to the batch version. Intuitively, this is because we require different bounds on error terms $E_k$ in the online and the batch settings. The batch setting considers an empirical estimate, hence operates on the averaged error. Whereas in the online setting we suffer from the per sample error. Efficient concentration bounds exist for the batch case [16], while for the online case, no such bounds exist in general. Hence, we conjecture that our bounds in Theorem 2 are *unimprovable* in the online setting.

**Approximation Error:** Note that the optimal decomposition $M^* = S^* + L^*$ is not identifiable in general without the incoherence-style conditions [15, 17]. In this paper, we provide efficient guarantees without assuming such strong incoherence constraints. This implies that there is an *approximation error* which is incurred even in the noiseless setting due to model non-identifiability.

**Algorithm 2:** Regularized Epoch-based Admm for Stochastic Opt. in high-dimensioN 2 (REASON 2)

---

**Input** $\rho, \rho_x$, epoch length $T_0$ , regularization parameters $\{\lambda_i, \mu_i\}_{i=1}^{k_T}$, initial prox centers $\tilde{S}_1, \tilde{L}_1$, initial radii $R_1, \tilde{R}_1$.

**Define** $Shrink_\kappa(a)$ shrinkage operator as in REASON 1, $G_{M_k} = M_{k+1} - S_k - L_k - \frac{1}{\rho}Z_k$.

**for** *each epoch* $i = 1, 2, ..., k_T$ **do**

Initialize $S_0 = \tilde{S}_i$, $L_0 = \tilde{L}_i$, $M_0 = S_0 + L_0$.

**for** *each iteration* $k = 0, 1, ..., T_0 - 1$ **do**

$$M_{k+1} = \frac{-\nabla f(M_k) + Z_k + \rho(S_k + L_k) + \rho_x M_k}{\rho + \rho_x}$$

$$S_{k+1} = \min_{\|S - \tilde{S}_i\|_1 \leq R_i} \lambda_i \|S\|_1 + \frac{\rho}{2\tau_k}\|S - (S_k + \tau_k G_{M_k})\|_{\mathbb{F}}^2$$

$$L_{k+1} = \min_{\|L - \tilde{L}_i\|_* \leq \tilde{R}_i} \mu_i \|L\|_* + \frac{\rho}{2}\|L - Y_k - U_k/\rho\|_{\mathbb{F}}^2$$

$$Y_{k+1} = \min_{\|Y\|_\infty \leq \alpha/p} \frac{\rho}{2\tau_k}\|Y - (L_k + \tau_k G_{M_k})\|_{\mathbb{F}}^2 + \frac{\rho}{2}\|L_{k+1} - Y - U_k/\rho\|_{\mathbb{F}}^2$$

$$Z_{k+1} = Z_k - \tau(M_{k+1} - (S_{k+1} + L_{k+1}))$$

$$U_{k+1} = U_k - \tau(L_{k+1} - Y_{k+1}).$$

**Set**: $\tilde{S}_{i+1} = \frac{1}{T_0}\sum_{k=0}^{T_0-1} S_k$ and $\tilde{L}_{i+1} := \frac{1}{T_0}\sum_{k=0}^{T_0-1} L_k$

**if** $R_i^2 > 2(s + r + \frac{(s+r)^2}{p\gamma^2})\frac{\alpha^2}{p}$ **then** Update $R_{i+1}^2 = R_i^2/2$, $\tilde{R}_{i+1}^2 = \tilde{R}_i^{\,2}/2$;

**else** STOP;

---

| Dimension | Run Time (s) | Method | error at 0.02T | error at 0.2T | error at T |
|---|---|---|---|---|---|
| | | ST-ADMM | 1.022 | 1.002 | 0.996 |
| d=20000 | T=50 | RADAR | 0.116 | 2.10e-03 | 6.26e-05 |
| | | REASON | 1.5e-03 | 2.20e-04 | 1.07e-08 |
| | | ST-ADMM | 0.794 | 0.380 | 0.348 |
| d=2000 | T=5 | RADAR | 0.103 | 4.80e-03 | 1.53e-04 |
| | | REASON | 0.001 | 2.26e-04 | 1.58e-08 |
| | | ST-ADMM | 0.212 | 0.092 | 0.033 |
| d=20 | T=0.2 | RADAR | 0.531 | 4.70e-03 | 4.91e-04 |
| | | REASON | 0.100 | 2.02e-04 | 1.09e-08 |

Table 3: *Least square regression problem, epoch size* $T_i = 2000$*, Error*$= \frac{\|\theta - \theta^*\|_2}{\|\theta^*\|_2}$.

Agarwal et al. [13] achieve an approximation error of $s\alpha^2/p^2$ for their batch algorithm. Our online algorithm has an approximation error of $\max\{s + r, p\}\alpha^2/p^2$, which is decaying with $p$. It is not clear if this bound can be improved by any other online algorithm.

### 3.2.1 Optimal Guarantees for Various Statistical Models

We now list some statistical models under which we achieve the batch-optimal rate for sparse+low rank decomposition.

**1) Independent Noise Model:** Assume we sample i.i.d. matrices $X_k = S^* + L^* + N_k$, where the noise $N_k$ has independent bounded sub-Gaussian entries with $\max_{i,j} \text{Var}(N_k(i,j)) = \sigma^2$. We consider the square loss function, $\|X_k - S - L\|_{\mathbb{F}}^2$. Hence $E_k = X_k - S^* - L^* = N_k$. From [Thm. 1.1][18], we have w.h.p. $\|N_k\| = \mathcal{O}(\sigma\sqrt{p})$. We match the batch bound in [13] in this setting. Moreover, Agarwal et al. [13] provide a minimax lower bound for this model, and we match it as well. Thus, we achieve the optimal convergence rate for online matrix decomposition for this model.

**2) Linear Bayesian Network:** Consider a $p$-dimensional vector $y = Ah + n$, where $h \in \mathbb{R}^r$ with $r \leq p$, and $n \in \mathbb{R}^p$. The variable $h$ is hidden, and $y$ is the observed variable. We assume that the vectors $h$ and $n$ are each zero-mean sub-Gaussian vectors with i.i.d entries, and are independent of

| Run Time | $T = 50$ sec | | | $T = 150$ sec | | |
|---|---|---|---|---|---|---|
| Error | $\frac{\|M^*-S-L\|_{\mathbb{F}}}{\|M^*\|_{\mathbb{F}}}$ | $\frac{\|S-S^*\|_{\mathbb{F}}}{\|S^*\|_{\mathbb{F}}}$ | $\frac{\|L^*-L\|_{\mathbb{F}}}{\|L^*\|_{\mathbb{F}}}$ | $\frac{\|M^*-S-L\|_{\mathbb{F}}}{\|M^*\|_{\mathbb{F}}}$ | $\frac{\|S-S^*\|_{\mathbb{F}}}{\|S^*\|_{\mathbb{F}}}$ | $\frac{\|L^*-L\|_{\mathbb{F}}}{\|L^*\|_{\mathbb{F}}}$ |
| REASON 2 | 2.20e-03 | 0.004 | 0.01 | 5.55e-05 | 1.50e-04 | 3.25e-04 |
| IALM | 5.11e-05 | 0.12 | 0.27 | 8.76e-09 | 0.12 | 0.27 |

Table 4: *REASON 2 and inexact ALM, matrix decomposition problem.* $p = 2000$, $\eta^2 = 0.01$

one another. Let $\sigma_h^2$ and $\sigma_n^2$ be the variances for the entries of $h$ and $n$ respectively. Without loss of generality, we assume that the columns of $A$ are normalized, as we can always rescale $A$ and $\sigma_h$ appropriately to obtain the same model. Let $\Sigma_{y,y}^*$ be the true covariance matrix of $y$. From the independence assumptions, we have $\Sigma_{y,y}^* = S^* + L^*$, where $S^* = \sigma_n^2 I$ is a diagonal matrix and $L^* = \sigma_h^2 AA^\top$ has rank at most $r$.

In each step $k$, we obtain a sample $y_k$ from the Bayesian network. For the square loss function $f$, we have the error $E_k = y_k y_k^\top - \Sigma_{y,y}^*$. Applying [Cor. 5.50][19], we have, with w.h.p. $\|n_k n_k^\top - \sigma_n^2 I\|_2 = \mathcal{O}(\sqrt{p}\sigma_n^2)$, $\|h_k h_k^\top - \sigma_h^2 I\|_2 = \mathcal{O}(\sqrt{p}\sigma_h^2)$. We thus have with probability $1 - Te^{-cp}$, $\|E_k\|_2 \leq \mathcal{O}\left(\sqrt{p}(\|A\|^2\sigma_h^2 + \sigma_n^2)\right)$, $\forall k \leq T$. When $\|A\|_2$ is bounded, we obtain the optimal bound in Theorem 2, which matches the batch bound. If the entries of $A$ are *generically* drawn (e.g., from a Gaussian distribution), we have $\|A\|_2 = \mathcal{O}(1 + \sqrt{r/p})$. Moreover, such generic matrices $A$ are also "diffuse", and thus, the low rank matrix $L^*$ satisfies Assumption A5, with $\alpha \sim \text{polylog}(p)$. Intuitively, when $A$ is generically drawn, there are diffuse connections from hidden to observed variables, and we have efficient guarantees under this setting.

## 4 Experiments

**REASON 1:** For sparse optimization problem, we compare REASON 1 with RADAR and ST-ADMM under the least-squares regression setting. Samples $(x_t, y_t)$ are generated such that $x_t \in \text{Unif}[-B, B]$ and $y_t = \langle \theta^*, x \rangle + n_t$. $\theta^*$ is $s$-sparse with $s = \lceil \log d \rceil$. $n_t \sim \mathcal{N}(0, \eta^2)$. With $\eta^2 = 0.5$ in all cases. We consider $d = 20, 2000, 20000$ and $s = 1, 3, 5$ respectively.

The experiments are performed on a 2.5 GHz Intel Core i5 laptop with 8 GB RAM. See Table 3 for experiment results. It should be noted that RADAR is provided with information of $\theta^*$ for epoch design and recentering. In addition, both RADAR and REASON 1 have the same initial radius. Nevertheless, REASON 1 reaches better accuracy within the same run time even for small time frames. In addition, we compare relative error $\|\theta - \theta^*\|_2 / \|\theta^*\|_2$ in REASON 1 and ST-ADMM in the first epoch. We observe that in higher dimension error fluctuations for ADMM increases noticeably (see Figure 1). Therefore, projections of REASON 1 play an important role in denoising and obtaining good accuracy.

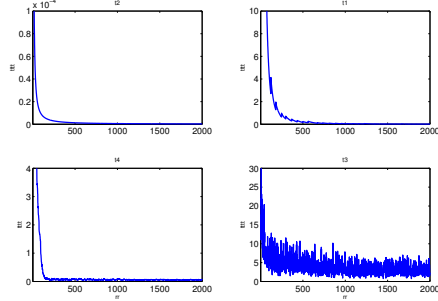

Figure 1: Least square regression, Error= $\frac{\|\theta-\theta^*\|_2}{\|\theta^*\|_2}$ vs. iteration number, $d_1 = 20$ and $d_2 = 20000$.

**REASON 2:** We compare REASON 2 with state-of-the-art inexact ALM method for matrix decomposition problem (ALM codes are downloaded from [20]). Table 4 shows that with equal time, inexact ALM reaches smaller $\frac{\|M^*-S-L\|_{\mathbb{F}}}{\|M^*\|_{\mathbb{F}}}$ error while in fact this does not provide a good decomposition. Further, REASON 2 reaches useful individual errors. Experiments with $\eta^2 \in [0.01, 1]$ show similar results. Similar experiments on exact ALM shows worse performance than inexact ALM.

**Acknowledgment**

We acknowledge detailed discussions with Majid Janzamin and thank him for valuable comments on sparse and low rank recovery. The authors thank Alekh Agarwal for detailed discussions of his work and the minimax bounds. A. Anandkumar is supported in part by Microsoft Faculty Fellowship, NSF Career award CCF-1254106, NSF Award CCF-1219234, and ARO YIP Award W911NF-13-1-0084.

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
