[Reviews · NeurIPS 2014]

Submitted by Assigned_Reviewer_4

The paper propose a multi-step version of stochastic ADMM [2], where it incorporate multiple steps or epochs and apply it to high dimensional problems. They apply it to two problems; (i) sparse optimization with a loss and a single regularizer, and (ii) matrix decomposition with a general loss function. For both problem settings the paper provides high dimensional guarantees under mild assumptions.

Originality and significance: The paper address an important problem of extending ADMM to high dimensional problems and provides theoretical guarantees. The results also looks encouraging. The paper builds up on the previous work of stochastic ADMM [2], but the contribution and improvement are significant.

Quality: The paper is strong on both theoretical and experimental aspects of the problem.

Clarity: It is well written and easy to follow.
Summary: The paper address an important problem of extending ADMM to high dimensional problems and provides theoretical guarantees. The experimental results also looks promising. It extends stochastic ADMM [2] significantly and clearly compares itself against the previous works.

Submitted by Assigned_Reviewer_25

This paper proposed a multistep version of Stochastic ADMM method for high-dimension problems, in which there are multiple epochs defined. In each epoch, projections onto certain norm balls with decreasing radius are employed.
As to the theoretical results, the authors showed that this simple variant achieves the minimax bound for sparse regression problem; while considering matrix decomposition problem, an excess risk bound, which is worse by a factor of d than the batch version, is obtained. Numerical simulation confirmed that the proposed algorithm performs better under both cases.

The work is a little incremental considering that the proposed method is the combination of ST-ADMM with the method proposed in [5]. Both algorithms have the same algorithmic framework. In fact, the O(s \log d /T) excess risk bound is not very surprising, because O(1/T) comes from the doubling trick used in [5] and an earlier work [**]
[**] Elad Hazan, Satyen Kale, Beyond the regret minimization barrier: an optimal algorithm for stochastic strongly-convex optimization, COLT’11
The factor s\logd is standard for high-dimensional problem with sparse structure.

Furthermore, the authors mistakenly use regret bound to refer to excess risk bound through the paper, which is misleading.

More surprisingly, the proposed method using stochastic ADMM outperforms [5] on sparse regression significantly in the empirical experiments, while they have the same order of excess rick bound. Note that sparse regression is a non-constrained problem in nature. Solving it using ADMM by introducing some auxiliary variable works better than directly solving the unconstrained problem using stochastic RDA is very counter-intuitive.

Also, the organization of this paper should also be improved for better understanding. It is better to present a unified algorithm, and then give two examples, rather than presenting two algorithms separately for sparse regression and matrix decomposition.

====
The authors explained why their algorithm outperforms multi-step RDA in the rebuttal. I have no concern about the empirical experiments any more.
Summary: The work is a little incremental considering existing work. Moreover, the empirical experiments delivered are a little counterintuitive.

Submitted by Assigned_Reviewer_30

Summary: The paper proposes a multi-step SADMM algorithm to address two regularized optimization problems, sparse optimization over vectors and extension to matrix decomposition, with a focus on high-dimensionality. They provide bounds on error for both problems in high probability under their stated assumptions. The authors claim that their sparse optimization method (with an adaptive epoch length) achieves minimax optimal error decaying rate, and that their matrix decomposition algorithm achieves optimal guarantees under two generative models.

Quality: The paper is clearly written and well-organized. The focus on high-dimensionality is clear and the proofs are well-developed. One drawback is that their experimental results are obtained over synthesized data instead of real-world data.

Clarity: The setting, assumptions, and notations are clearly stated in their paper. The comparisons between their algorithm and previous work are clear (both in terms of the theoretical bounds and the empirical evaluations).

Originality: The paper distinguishes itself by focusing on addressing regularized optimization problems with high-dimensionality, and providing theoretical guarantee under clear assumptions.

Significance: They proposed two algorithms that deal with high-dimensional sparse optimization problems for vectors and matrices. They develop theoretical guarantees under clear assumptions for both schemes, and obtain superior performance on synthesized data (which satisfy their model assumption). However, one concern is the lack of comparison on real-world data.
Summary: The paper focuses on solving high-dimensional sparse optimization problems for vectors and matrices via a new version of SADMM. They develop theoretical guarantees under clear assumptions for both schemes, and obtain superior performance on synthesized data. It has overall high quality.
Author Feedback
Author rebuttal: Reviewer 1:

Thank you for your comments. We will replace “regret bound” by “excess risk bound” throughout the paper.

The idea of using epochs was inspired by (Agarwal et al 2012), who themselves acknowledge a series of previous works which use these ideas. But our work is more than just applying epochs to an ADMM framework. The main contribution is the matrix decomposition problem and the framework can be extended to any multiblock problem and provide guarantees in such cases. The results on matrix decomposition are highly non-trivial since they require careful decoupling of errors for sparse and low rank parts. Note that Agarwal et al only handle the case with a single regularizer and do not have to address decoupling of errors. This is the first work to provide guarantees for online learning of multi-block systems with general decomposable regularizers in the high dimensional regime.

Note that in the matrix decomposition case, the excess regret bound is worse than batch by a factor of d (in paper notation, p) only in the worst-case scenario. If the noise is random enough (i.e. for independent noise model and linear Bayesian networks) and s=O(r), our bound matches the batch version in terms of d and this is an important contribution of the paper. Our excess risk bound is not an immediate result from any of the previous works.

As for simulation results, we are comparing REASON with RADAR. Both these methods constrain the answer within a norm ball. The goal is to find a sparse solution, while REASON directly uses the l1 norm; RADAR approximates the l1 ball with some lp ball for p close to 1. This results in better accuracy for REASON. We will add this reasoning to the paper.

Considering organization of the paper, our goal is to give some simple intuition first through sparse regression and then handle multiblock systems such as matrix decomposition, which is the main contribution.

Reviewer 2:
Thank you for your positive comments. In future we plan to run experiments on a large scale and on real datasets and the focus of the current work is to provide theoretical guarantees with validating experiments.

Reviewer 3:
Thank you for your positive and careful review.